# CBL/CAP Is Essential for Mitochondria Respiration Complex I Assembly and Bioenergetics Efficiency in Muscle Cells

**DOI:** 10.3390/ijms24043399

**Published:** 2023-02-08

**Authors:** Cho-Cho Aye, Dean E. Hammond, Sergio Rodriguez-Cuenca, Mary K. Doherty, Phillip D. Whitfield, Marie M. Phelan, Chenjing Yang, Rafael Perez-Perez, Xiaoxin Li, Angels Diaz-Ramos, Gopal Peddinti, Matej Oresic, Antonio Vidal-Puig, Antonio Zorzano, Cristina Ugalde, Silvia Mora

**Affiliations:** 1The Department of Cellular and Molecular Physiology, Institute of Translational Medicine, The University of Liverpool, Crown Street, Liverpool L69 3BX, UK; 2Metabolic Research Laboratories, Wellcome Trust-MRC Institute of Metabolic Science, Addenbrooke’s Hospital, University of Cambridge, Cambridge CB2 0QQ, UK; 3Division of Biomedical Sciences, Centre for Health Science, University of the Highlands and Islands, Old Perth Road, Inverness IV2 3JH, UK; 4Glasgow Polyomics, College of Medical, Veterinary and Life Sciences, Garscube Campus, University of Glasgow, Glasgow G61 1BD, UK; 5Centre for Nuclear Magnetic Resonance, Institute of Integrative Biology, University of Liverpool, Crown Street, Liverpool L69 3BX, UK; 6Instituto de Investigación, Hospital Universitario 12 de Octubre, Avda. de Córdoba s/n, 28041 Madrid, Spain; 7Centro de Investigación Biomédica en Red de Enfermedades Raras (CIBERER), U723, 28029 Madrid, Spain; 8Institute for Research in Biomedicine, C/Baldiri Reixac 10, 08028 Barcelona, Spain; 9Technical Research Centre of Finland, 02044 Espoo, Finland; 10Turku Centre for Biotechnology, University of Turku and Abo Akademi University, 20520 Turku, Finland; 11School of Medical Sciences, Örebro University, 702 81 Örebro, Sweden; 12Wellcome Trust Sanger Institute, Hinxton, Cambridge CB10 1SA, UK; 13CIBER de Diabetes y Enfermedades Metabólicas Asociadas (CIBERDEM), Instituto de Salud Carlos III, 28029 Madrid, Spain; 14Department de Bioquimica i Biomedicina, Facultat de Biologia, Universitat de Barcelona, 08028 Barcelona, Spain; 15Institut de Biomedicina de la Universitat de Barcelona (IBUB), Universitat de Barcelona, 08028 Barcelona, Spain

**Keywords:** CBL, SORBS1, mitochondria, insulin signalling, insulin resistance, glucose transport

## Abstract

CBL is rapidly phosphorylated upon insulin receptor activation. Mice whole body CBL depletion improved insulin sensitivity and glucose clearance; however, the precise mechanisms remain unknown. We depleted either CBL or its associated protein SORBS1/CAP independently in myocytes and assessed mitochondrial function and metabolism compared to control cells. CBL- and CAP-depleted cells showed increased mitochondrial mass with greater proton leak. Mitochondrial respiratory complex I activity and assembly into respirasomes were reduced. Proteome profiling revealed alterations in proteins involved in glycolysis and fatty acid degradation. Our findings demonstrate CBL/CAP pathway couples insulin signaling to efficient mitochondrial respiratory function and metabolism in muscle.

## 1. Introduction

Increasing evidence suggests insulin signaling is important for mitochondrial function: bioinformatic approaches identified genes connecting insulin signaling with mitochondria [1]; insulin and IGF1 regulate expression of mitochondrial respiratory protein components [2,3,4]; and insulin administration to healthy humans increases ATP production and the expression of mitochondrial protein transcripts [5]. In contrast, insulin deficiency in humans [6] and in mice [3,7] leads to lower mitochondrial respiration and ATP production. Individuals with insulin resistance exhibit reduced expression of mitochondrial proteins [8,9,10,11,12]. Decreased mitochondrial DNA, oxidative enzyme activity [9] and mitochondrial size and density [9,11,13,14] have been reported in diabetes. Other studies showed impaired mitochondrial respiration in lean insulin-resistant offspring of type 2 diabetic patients [15,16,17,18,19], although this is controversial [20,21].

Insulin binding to its receptor activates multiple signaling cascades that are responsible for the pleiotropic effects of the hormone in different tissues. Different intracellular adaptor proteins are recruited and mediate these specific effects. Casitas b-lineage lymphoma (c-CBL, hereafter referred as CBL), one of these proteins, becomes rapidly phosphorylated following insulin administration [22]. In adipocytes, CBL is associated intracellularly with another protein referred to as CAP (for CBL Associated Protein), also known as SORBS1 [23,24] due to its Sorbin homology domain. Upon the phosphorylation of CBL, the CBL/CAP complex provides docking sites for the further recruitment and activation of downstream signaling molecules that facilitate insulin-stimulated glucose uptake through cytoskeletal rearrangements [25].

The importance and mechanism of action of the CBL/CAP pathway in muscle is less clear, although several polymorphisms in SORBS1 (CAP) have been associated with the development of diabetes in humans [23,26]. We reported that CBL phosphorylation in response to insulin in muscle does not activate the same signaling cascades as in adipocytes [22]. This suggests CBL may play a different role in muscle. Furthermore, whole body CBL depletion in mice resulted in a paradoxical increased glucose clearance and insulin sensitivity [27]. The skeletal muscle of these animals exhibited increased AMP-regulated kinase activation. A similar phenotype was seen in a knock-in animal model expressing a mutated CBL [28]. However, the molecular mechanisms remain unknown.

To shed light on this mechanism and understand CBL function in muscle, we depleted CBL or CAP expression independently in the C2C12 cell line by lentiviral-mediated stable expression of specific shRNAs and compared the effects to cells expressing non-targeting control shRNAs. We show that CBL or CAP depletion in myocytes reduces mitochondrial respiratory chain complex I activity and compromises the assembly of respiratory proteins into supercomplexes. We provide evidence these impairments lead to a unsuccessful compensatory response aiming to increase the number of mitochondria but which causes derangements in the cellular proteome and metabolism.

## 2. Results

### 2.1. Cbl Depletion in Myocytes Decreases ATP Levels and Enhances AMPK Activity, Glucose Transport, and Mitochondrial Mass

To gain further insights onto the molecular mechanisms of the CBL-mediated signaling pathway in skeletal muscle cells, we depleted CBL in C2C12 myocytes. As a complementary experimental approach, we deleted CAP, a CBL-binding protein found to be required for CBL recruitment into the proximity of the insulin receptor and important for CBL signaling. Both proteins were reduced by lentiviral-mediated stable expression of validated shRNAs (Figure 1A) in independent sets of cells. Untransfected cells and cells expressing a scrambled non-targeting shRNA (NT-shRNA) construct were used as controls (Figure 1A). NT-shRNA expressing cells did not show any changes in CBL or CAP expression, while cells expressing CBL- or CAP-specific shRNAs demonstrated efficient knockdown of CBL and CAP, respectively (Figure 1A).

The skeletal muscle of global CBL-null mice display increased AMP-regulated kinase (AMPK) activation [27]. Similarly, CBL KD cells exhibited elevated AMPK phosphorylation levels at Thr172, suggesting a primary autonomous-cell-dependent increase in AMPK activation even in basal (non-insulin stimulated) state (Figure 1B). Basal AMPK phosphorylation levels in both cells stably expressing NTshRNA or shRNAs for CAP, were similar to untransfected control cells. The phosphorylation levels of AMPK increased across all the experimental groups following insulin treatment (Figure 1B). Consistent with increased basal AMPK activation, CBL-depleted cells had significantly lower ATP levels compared to other groups. Conversely, cells expressing shRNAs for CAP or NTshRNA displayed similar cellular ATP levels to untransfected cells (Figure 1C). CBL KD and CAP KD cells exhibited activation of phosphatidylinositol 3-kinase and of p44/p42 extracellular regulated kinases (ERK) following insulin treatment, with profiles comparable to those observed in control untransfected cells (Appendix A). However, CBL- and CAP-KD cells had increased ERK activation at earlier insulin time points (5 min post-stimulation). Additionally, NTshRNA and CAP KD had higher basal ERK activation than wild type untransfected cells. These effects were not related to differences in the expression of insulin receptor (Appendix A).

One of the expected physiological consequences of AMPK activation is an increase in glucose transport. We assessed basal and insulin-stimulated 2-deoxyglucose transport in differentiated control myocytes or those depleted of CBL or CAP. Basal glucose transport in CBL KD and CAP KD cells was significantly elevated compared to control cells. Insulin-mediated glucose transport was also higher in CBL KD cells (Figure 1D). Interestingly, the depletion of CBL or CAP in 3T3L1 adipocytes did not change ATP cellular levels compared to control cells (Figure 1E).

PGC1α and PGC1β are two transcription coactivators that regulate mitochondrial biogenesis and are regulated by AMPK. We measured their expression and that of NRF1 and TFAM by qPCR. *Pgc1α* and *Pgc1β* mRNA levels were significantly increased in CBL- and CAP-depleted cells, whereas no statistical differences were seen for *Nrf1* and *Tfam* (Figure 2A). Next, we sought to determine whether there was an increase in mitochondrial mass. To this end, we quantified mtDNA by qPCR and quantitated mitochondria stained with JC-1. These experiments showed that mitochondrial mass was elevated in CBL- and CAP-depleted cells compared to control cells (Figure 2B).

### 2.2. CBL- and CAP-Depleted Myocytes Show Alterations in Mitochondrial Respiration

The reduced ATP content and increased AMPK activity in CBL KD cells, together with a paradoxical increase in mitochondrial mass, suggested a compensatory mechanism [29] in response to an overall cellular bioenergetics failure secondary to defective CBL. We posited that in this context, the increase in ADP/ATP ratio reflective of decreased energy availability may reflect an impaired energy production in the mitochondria. To assess the mitochondrial function in these cells, we first determined the rate of mitochondrial respiration (OCR) in CBL or CAP KD cells in comparison to control cells. Basal cellular OCR was not different in CBL KD or CAP KD cells compared to control cells (Figure 3A). However, CBL- and CAP-KD cells exhibited higher rates of mitochondrial respiration after inhibition with oligomycin, suggesting enhanced proton leak. Higher maximal respiratory capacity (in the presence of the uncoupler FCCP) was reduced in CBL KD cells compared to control cells (Figure 3A). We did not detect changes in the extracellular acidification rate (ECAR) (Figure 3B) or in the abundance of reactive oxygen species in the steady state compared to control cells (Figure 3C).

The reconciliation of these data with the increase in mitochondria mass observed in CBL- and CAP-KD cells required mitochondria in CBL KD cells to be less active or efficient than those in control cells. To test this hypothesis, we next determined the activities of the mitochondrial respiratory chain complexes in vitro, each under non-limiting substrate availability. We found that the activity of OXPHOS complex I (CI) was significantly reduced in CBL KD and CAP KD cells compared to control cells (Figure 4A), whereas no significant changes were observed in the activities of the remaining respiratory chain complexes (Figure 4B,C). In agreement with low ATP levels, ATP synthase activity (complex V) was also significantly reduced in CBL KD cells (Figure 4D) compared to control cells.

### 2.3. Reduced Mitochondrial Respiratory CI Expression and Respirasome Assembly in CBL Cells

The above experiments indicated that CBL- and CAP-KD cells had impaired mitochondrial function. Accumulated evidence in several organisms has demonstrated that respiratory chain complexes I, III, and IV are arranged in supramolecular assemblies known as supercomplexes (SCs) or respirasomes. These are thought to enhance the efficiency of OXPHOS function, restrain ROS production, or promote the stability of the respiratory complexes [30,31]. Mammalian complexes I, III_2_, and IV assemble into SCs I + III_2_, III_2_ + IV_1_, or I + III_2_ + IV_1–4_, with ATP synthases (CV) forming dimers. Complex II (CII) is the only complex, which does not associate with other respiratory complexes, possibly because it also participates in the TCA cycle.

To determine whether the impairment in CI enzyme activity was due to alterations in the structural organization of this complex and SCs, we used blue-native polyacrylamide gel electrophoresis (BN-PAGE), a method that allows the separation of the large native complexes following the mild solubilization of mitochondrial membranes with non-ionic detergents. We examined the formation of SCs in the steady state of CBL KD, CAP KD, and control myocytes by Western blotting. Equal protein levels of each mitochondrial fraction were used to load the BN-PAGE and data were normalized to the content of SHDA (a complex II subunit). While the relative abundance of NDUFA9 (a complex I subunit), CORE2 (a complex III subunit), and COX5A (a complex IV subunit) relative to SDHA (CII) in both CBL- and CAP-KD cells was marginally lower than in control cells, this was not statistically significant. However, the quantification of our BN-PAGE experiments showed that CBL- and CAP-KD cells have a significantly lower abundance of the respirasomes (SCs I+III+IV) compared to control cells (Figure 5), which was not attributable to the SC assembly factor COX7A2L, whose levels were normal (Appendix A). These results indicate CBL- and CAP-KD cells had a specific defect in CI.

### 2.4. Cbl KD Cells Have Reduced Levels of Proteins Involved in CI Biogenesis and Fatty Acid Degradation, and Increased Levels of Glycolytic Enzymes

To dissect further the molecular mechanisms dysregulated by CBL depletion, we performed quantitative SILAC proteomics on mitochondrial preparations from CBL KD and untransfected control cells. In parallel, we compared the proteome of NTshRNA cells to untransfected cells. We detected 275 differentially expressed proteins in total. Reactome and KEGG pathway enrichment analysis of the 39 proteins downregulated by ≥1.5 fold change following CBL knockdown revealed an overrepresentation of terms linked with the TCA, respiratory electron transport, and mitochondrial fatty acid oxidation. In contrast, the 62 proteins that were upregulated by ≥1.5 fold were associated with carbohydrate metabolism, carbon metabolism, and extracellular matrix organization (Figure 6A). A diagram depicting the ≥1.5 fold differentially expressed proteins in CBL KD cells compared to control is shown in Figure 6B, and the equivalent comparison between NTshRNA and control untransfected cells is shown in Appendix A. Notably, CBL KD cells had significantly lower levels of the CI core subunit NDUFS2 (Figure 6B), which contains a cofactor binding site for a [4Fe-4S] cluster essential for CI activity, as well as for binding the assembly factors required for CI biogenesis, AIFM1 and ECSIT. No changes in NDFUS2 were detected in cells expressing the NT-shRNA (Appendix A).

Consistent with the increase in glucose uptake data, CBL KD cells exhibited increased expression of several glycolytic enzymes, specifically pyruvate kinase (PKM1), glucose 6-phosphate isomerase (GPI), enolase (ENO1) and 6-phosphogluconate dehydrogenase (PGD), involved in the production of ribulose 5-phosphate in the pentose phosphate pathway, a main generator of cellular NADPH (Figure 6B).

We detected lower expression of proteins involved in fatty acid oxidation (Figure 6B), including long-chain-specific acyl-CoA dehydrogenase (ACADL), acetyl CoA acetyltransferase (ACAA2), and the alpha and beta subunits of the trifunctional enzyme (HADHB), an inner mitochondrial membrane protein complex involved in fatty acid oxidation that also interacts with CI [32] and has been suggested to facilitate substrate channeling. We also found lower levels of the mitochondrial calcium uniporter (MICU2) and mitochondrial import receptor subunit TOMM40 (Figure 6B).

Interestingly, protein processing in the ER was elevated in our analysis, and we found higher levels of the heat shock protein 90 chaperone machinery (HSP90AA1 and HSP90AB1) (Figure 6B), a key regulator of proteostasis under both physiological and stress conditions.

To confirm these metabolic derangements, we analyzed the metabolome of cellular lysates of CBL KD, CAP KD and control cells by ^1^H-NMR. Consistent with the proteomic data, metabolite analyses showed an accumulation of essential amino acids, including branched chain amino acids, suggesting increased proteolysis in CBL- and CAP-deficient cells and a lower abundance of TCA intermediaries (Appendix A).

### 2.5. CBL- and CAP-KD Cells Have Reduced Expression of Genes Involved in Fatty Acyl-CoA Import into Mitochondria

AMPK activation in muscle increases fatty acid oxidation by facilitating the incorporation of fatty acyl CoAs into the mitochondria. However, our proteomic data detected lower expression of several proteins involved in fatty acid oxidation in CBL KD cells. To confirm this, we determined the expression of carnitine palmitoyl transferase-1 (CPT1) the protein that facilitates the import of fatty acyl-CoAs in the mitochondria and of long chain acyl-CoA lyases (ACSL) and determined the lipid content in CBL- and CAP-KD cells by mass spectrometry. We found that mRNA levels of CPT1 and of several ACSLs isoforms were significantly downregulated in both CBL- and CAP-KD cells (Figure 7A,B), concomitantly with significantly lower in vitro ACSL activity on ^14^C-palmitate (Figure 7C), indeed suggesting lower fatty acid utilization.

Furthermore, lipid content analysis of CBL KD, CAP KD, and control cells (Appendix A), revealed that total triglyceride content was not different among the different sample groups, but triglycerides in CBL KD cells were more unsaturated and shorter fatty acids. CBL KD cells had decreased levels of ceramides, lysophosphatydil ethanolamine, and lyspophosphatydil choline compared to control cells, whereas CAP-depleted cells had increased etherlipids (Appendix A).

## 3. Discussion

We provide experimental evidence of a novel direct link between insulin signaling and the regulation of mitochondrial function and metabolism in muscle. We demonstrate that CBL and CAP are required for efficient mitochondrial respiratory complex function in muscle cells. We and others have previously reported that CBL null mice are more glucose tolerant [27,28,33] as they over activate AMP-regulated kinase in skeletal muscle [27].

Here we show these are autonomous tissue effects fully recapitulated in C2C12 muscle cells depleted of CBL through shRNA-mediated knockdown. Moreover, our data define the molecular mechanism by which CBL deficiency increases AMPK activation in muscle.

We show that either CBL or CAP deficiency compromises maximal respiratory capacity through the inhibition of mitochondrial respiratory chain CI enzyme activity and decreased respiratory supercomplex formation. These defects limit ATP production and activate AMPK and the mitochondrial biogenesis program, resulting in higher numbers of aberrant mitochondria. We show CBL deficiency leads to major alterations in the cellular proteome, affecting pathways related to CI biogenesis and function, glycolysis, and fatty acid degradation.

The effector mechanisms derived from the genetic ablation of CBL involve the activation of AMP kinase, an evolutionarily conserved serine threonine kinase that functions as an energy sensor to preserve energy availability [34]. AMPK promotes mitochondrial biogenesis through the phosphorylation and activation of PGC-1α. CBL- and CAP-depleted myocytes showed elevated PGC-1α and PGC-1β expression and increased mitochondrial mass, as previously observed in the muscle of CBL null mice [27]. It is of pathophysiological relevance that CBL deficiency decreased ATP levels and activated AMPK specifically in muscle cells but not in differentiated 3T3L1 adipocytes, thus reinforcing the concept of tissue-specific selective CBL signaling in adipose and muscle cells [22].

With this increase in mitochondrial mass, CBL- and CAP-depleted cells grew and differentiated normally in glucose media and were able to maintain normal basal oxygen consumption similar to that of control cells, despite the substantially lower CI activity. CBL depletion resulted in a greater reduction in CI activity than CAP KD, and a concomitant reduction in the maximal respiratory capacity without increase in ROS generation. A similar phenotype has been observed in fibroblasts from patients harboring mutations in subunits NDUFA1 and NDUFV1 of CI [35].

We dissected potential scenarios that could explain the impairment of CI activity. We discovered that CBL-deficient cells had a significantly reduced content of NDUFS2, one of the 14 evolutionarily conserved core subunits of CI displaying catalytic activity and which are also important for the structural stability of this complex, as it binds assembly factors AIFM1 and ECSIT required for CI biogenesis [36]. Mutations in NDUFS2 cause severe reductions in CI activity and perturb the formation of CI assembly in patients with Leigh Syndrome [37]. These cells showed reduced levels of the CI assembly factors ECSIT and AIFM1, which also contribute to CI activity by regulating the assembly of the complex.

The precise mechanism by which CBL maintains NDUFS2 expression remains unknown. Since CBL is a multiadaptor signaling protein with many protein interactions, and in addition, it can function as an ubiquitin E3-ligase that transfers ubiquitin from an E2 ligase to target proteins for degradation, we may speculate that CBL could act by either activating a signaling cascade that converges to regulate transcription factor(s) responsible for NDUFAS2 expression in muscle, or alternatively, that CBL may target a negative regulator of NDUFAS2 through its E3-ligase activity. In this regard, it was shown by Molero et al. [28], that a heterozygous mice model expressing CBL protein with a loss-of-function mutation within the RING finger domain also exhibited increased AMPK activation and the mice were (like the CBL-null animals) more sensitive to insulin. These data suggest the mechanism for NDUFAS2 regulation by CBL may be related to its E3-ligase activity, but further experiments are required to identify the molecules involved.

CBL- and CAP-deficient cells also showed a decrease in the formation of the respirasome or SC I + III + IV without evident alterations in the levels of individual CII, CIII, and CIV. It has been suggested that the association of respiratory chain complexes in supercomplexes would provide functional advantages, such as the prevention of destabilization and degradation, the enhancement of electron transport efficiency [38], or a decrease in proton leak [39], thus facilitating efficient oxidation of the available substrates and cellular adaptation to energy demands. In this context, reduced SC formation may negatively impact the functional activity of the respiratory chain complexes, and thus could further contribute to the reduced CI activity seen in CBL- and CAP-depleted cells. While the proteins that mediate the assembly of all these complexes are not completely characterized, several groups identified COX7A2L- or COX7A-related protein (COX7RP), also known as SCAFI (for supercomplex assembly factor 1), as a protein that regulates supercomplex assembly [38,40,41]. We did not find, however, any significant differences in the expression of COX7A2L in CBL- or CAP-KD cells, thus indicating that the decrease in SC I + III + IV formation is not attributable to COX7A2L but most probably due to a severe defect in CI biogenesis.

Our findings indicate CBL/CAP deficiencies trigger derangements in mitochondrial complex biogenesis and organization and prevent optimal CI activity and ATP production. Our data are consistent with literature reporting impaired insulin signaling in muscle, such as that observed in the insulin receptor null mice (MIRKO) or resulting from streptozotocin administration, leading to lower mitochondrial respiration and ATP production [3,7], since insulin deficiency and insulin resistance impair CBL/CAP signaling in mice [22]. Similar to our findings, Zabielski et al. noted decreased CI activity and downregulation of OXPHOS TCA and β-oxidation proteins [7]. Reduced mitochondrial electron transport has been reported in diabetic patients [42,43], and a recent study found reduced oxidative phosphorylation and SC assembly in the skeletal muscle of obese type 2 diabetic women [44]. All these findings are consistent with the data reported here.

AMPK activation in muscle potentiates fatty acid oxidation, facilitating the entrance of fatty acyl-CoAs into the mitochondria. However, CBL- and CAP-deficient cells had reduced ACSL activity and levels of mRNAs coding for long chain acyl-CoA liases and CPT1. Lower ACSL1 activity negatively impacts the synthesis of phospholipids and cardiolipins [45], which are also important for SC formation and mitochondrial function [46], and could contribute, at least in part, to the observed mitochondrial dysfunction. CBL-deficient cells had reduced lysoPE and lysoPC levels consistent with lower ACSL activity. Our findings of a reduced content of proteins involved in fatty acid oxidation and increased glycolysis in CBL-depleted cells is in agreement with findings in the muscle of STZ-induced diabetic mice [7].Triglyceride levels in CBL/CAP-deficient cells were, however, not different than control cells, in agreement with findings in mice lacking insulin receptor (MIRKO) or following streptozotocin-induced diabetes [3].

To summarize, we show CBL/CAP are required for the efficient functioning of mitochondrial respiratory complexes and energy metabolism in muscle cells, thus providing a novel link between insulin signaling and mitochondrial function. Impairments in CBL shift the cellular metabolism towards glucose utilization, thus explaining the enhanced glucose clearance seen in the CBL-null mice.

## 4. Materials and Methods

### 4.1. Materials and Antibodies

Reagents were purchased from Merck-Life Sciences (Gillingham, UK) or Thermo Fisher Scientific (Loughborough, UK). Primary antibodies for: phosphor-AMPK (T172), AMPK, phospho-AKT (Ser473), AKT, phosphop44/42 (ERK1/2) (Thr202/Tyr204), ERK1/2,c-CBL, CAP, CBL-b, were from Cell Signaling (Leiden, NL); β-TUBULIN and ACTIN from Merck-Sigma-Aldrich; NDUFA from Thermo Fisher; CORE2, COX5A and SDHA from Abcam. HRP-conjugated secondary antibodies were from Thermo Fisher.

### 4.2. Cell Culture and Knockdowns

C2C12 myocytes were obtained from ATCC (Manassas, VA, USA) and were mycoplasma-free. Cells were grown and differentiated as previously described [47]. Myotubes between days 3 and 5 following induction of differentiation were used for experiments. To generate c-CBL- or CAP-depleted cell lines, myoblasts were infected with a mixture of 4 lentiviral vectors, each expressing validated shRNA for either c-CBL or CAP (mission shRNA Merck Sigma-Aldrich). A control group was infected with viral particles expressing a non-targeting-shRNA. shRNA-expressing cells were selected in puromycin (2.5 µg/mL).

### 4.3. Cell Lysates and Western Blotting

Cell lysates and Western blotting were performed as described [22].

### 4.4. mRNA Extraction and qPCR

mRNA extraction and qPCR were performed as described [47]. Primer sequences are provided in the electronic Appendix A

### 4.5. Mitochondrial Mass

Extraction and quantification of mtDNA: qPCR assay to amplify a 162-nt region of the mitochondrial NADH dehydrogenase subunit 4 (*MT-ND4*) gene was adapted from Koekemoer et al. [48]. Fluorescence mitochondrial labelling: cells were stained with 10µg/mL of JC-1 (Life Technologies) for 10 min at 37 °C as previously described [47]. Emission fluorescence values were determined on 10,000 cells per condition in each experiment and analyzed with BD CellQuest^TM^Pro Software 5.2 (BD Biosciences, Wokingham, UK)).

### 4.6. Mitochondrial Isolation

Cells were washed with PBS and homogenized in a Dounce homogenizer in an isolation buffer containing: 2.5% BSA *w*/*v*, 220 nM mannitol, 70 mM sucrose, 1 mM EDTA, 10 mM Tris-HCl pH = 7.4. Lysates were centrifuged twice (700× *g*, 5 min, 4 °C) and the supernatants further centrifuged at 7000× *g*, 10 min, 4 °C. The pellets were resuspended in the isolation buffer and the centrifugation step was repeated twice. The pellet containing crude mitochondria was resuspended in 20 µL of buffer and used in further biochemical assays.

### 4.7. Cellular ATP Content

ATP content was determined with a luminescence kit from Merck-Sigma-Aldrich (catalogue FLAAM) according to manufacturer’s instructions as described in [47]. Briefly, cell lysates were prepared in 1% (*w*/*v*) TCA (trichloroacetic acid) and incubated for 15 min before centrifugation (11,300× *g*, 10 min, 4 °C). The supernatant containing ATP was neutralized by adding Tris/HCl buffer (50 μL, 0.1 M, pH 9.0) before it was mixed with the bioluminescent solution and read by a luminometer (Berthold Tube Master (Berthold Technologies, Harpenden, UK). ATP concentrations in cell lysates were then determined from a standard curve, and then normalized for the total protein concentrations.

### 4.8. Reactive Oxygen Species

Cells were stained with 2 µM DCF-DA (Life Technologies, Paisley, UK) at 37 °C in the dark for 30 min as previously described [47]. Average fluorescence values (arbitrary units) were calculated from 3 independent experiments, analyzing between *n* = 126 and 192 cells per group.

### 4.9. Oxygen Consumption (OCR) and Extracellular Acidification Rate (ECAR)

Cell respirometry was performed in a Seahorse Bioscience XF24 (Seahorse Bioscience, North Billerica, MS, USA) as described in [47]. Values were normalized to protein content.

### 4.10. Acyl-CoA Synthase Activity

Acyl-CoA synthase activity was measured in 10–30 µg of cellular lysates from differentiated cells using 50 µM of 1-^14^C-palmitate (specific activity 0.1 µCi/µL) for 5 min at 37 °C as described in [45]. The reaction was terminated by the addition of 1ml of Dole’s reagent and centrifugation at 1000× *g* for 3 min. The lower layer was washed twice in 2 mL of heptane and counted in 4 mL of scintillation cocktail. Data were normalized to protein content.

### 4.11. Activity of Mitochondrial Respiratory Complexes

Activity of mitochondrial respiratory complexes were determined as previously reported in [49,50]. We used cells grown in 10 cm dishes. Cells were trypsinized, pelleted, and resuspended in ice-cold phosphate buffered saline twice. The pellet was resuspended in buffer A (20 mM MOPS KOH pH = 7.4, 250 mM sucrose) and one volume of digitonin 0.2 mg/mL was added. Samples were then incubated in ice for 5 min, centrifuged at 5000× *g* for 3 min, and the supernatant discarded. The pellet was resuspended in 600 µL of buffer B (20 mM MOPS KOH pH = 7.4, 250 mM sucrose, sodium EDTA 1 mM), incubated in ice for 5 min, and centrifuged at 100,000× *g* for 3 min. The pellet was kept for the assays or aliquoted and stored at −80 °C. The pellets were resuspended in 200 µL of potassium phosphate buffer 10 mM, pH = 7.4, and the suspension frozen/thawed in liquid nitrogen/37 °C 3× times to disrupt mitochondrial membranes. An aliquot was then taken to measure protein content. For these assays, we used 50 µg of protein. To measure complex I activity, we measured the rotenone-sensitive NADH decylubiquinone oxidoreductase activity by monitoring the decline in absorbance of the NADH at 340 nm for 2 min. A total of 50 μg of sample was combined with buffer containing 20 mM potassium phosphate pH = 8.0, 0.2 mM NADH, 1 mM NaN_3_, 2.5 mg/mL BSA (prepared in EDTA 10 mM pH = 7.4) in a 1 mL spectrophotometric cuvette and preincubated for 2 min at 30 °C. Following this, the basal absorbance was registered at 340 nm for 2 min (Reading1). The reaction was started by the addition of decylubiquinone (Merk-Sigma catalogue number C7956) to a final concentration of 50 μM. The activity was measured by the following the decrease in absorbance at 340 nm resulting from the oxidation of NADH for 2 min (reading2). Following this, rotenone was added to a final concentration of 5 μM and the absorbance monitored for another 2 min. (reading 3). Calculation of Complex I activity (nmol/min*mg protein): (reading2-reading3)*1000*total volume (mL)/ε(=6.81 mL/nmol*cm)*sample volume (mL)* Protein concentration at mg/mL*1 cm (light path length).

Complex II + CoQ10 + III succinate cytochrome C oxidoreductase: for this, we measured the increase in absorbance at 550 nm, following the reduction in cytochrome C. The samples were preincubated for 30 min at 30 °C in a buffer containing phosphate buffer 45 mM, pH = 7.4 and 17 mM succinate, pH = 7.4. After this, NaN3 rotenone and cytochrome C were added to final concentrations: 4 mM NaN_3_, 7 μM rotenone, 0.065% (p/v) cytochrome C. The absorbance at 550 nm was measured for 5 min. Calculations: (nmol/min*mg protein): ΔAbsorbance_(550nm)_*1000 *total volume (mL)/ε(=21.1 mL/nmol*cm)* sample volume(mL)*protein concentration in mg/mL *1 cm light path length.

Complex IV (cytochrome C oxidase) was measured monitoring the oxidation of cytochrome C at 38 °C as the decrease in absorbance at 550 nm for 2 min in 1ml cuvettes. First reduced cytochrome C (Sigma C2506) was prepared as follows: cytochrome C was dissolved at 10 mg/mL concentration in 10% potassium phosphate buffer 0.1M pH = 7.0, with a pinch of sodium dithionite (Na_2_S2O_4_). Reduction in cytochrome C was measured at 550 nm, and should be around 2.0 ± 0.1. For the kinetic measurements, the buffer containing 10 mM phosphate buffer pH = 7.0, 1 mg/mL BSA (prepared in EDTA 10 mM, pH = 7.4), reduced cytochrome C 80 μM, and water was preincubated at 38 °C for 2 min prior to the addition of the sample (50 μg). The absorbance was then monitored at 550 nm for 2 min. Calculations: ΔAbsorbance(_550nm_)*total volume (mL)/ε(=21 mL/nmol*cm)* sample volume (mL)*protein concentration in mg/mL *1 cm.

Complex V (ATP synthase) activity was measured by monitoring the variation in absorbance at 340 nm following the oxidation of NADH for 2 min. A buffer was prepared containing 10 mM MgSO_4_ in Hepes KOH 100 mM pH = 8.0, NADH 0.3 mM, phosphoenolpyruvate 2.5 mM, pyruvate kinase 50 μg/mL, lactate dehydrogenase 50 μg/mL, antimycin A (prepared in ethanol 50%), at 2 μg/mL, homogenate volume for 50 ug protein and water to 1 mL. Basal readings were recorded at 340 nm for 2 min at 30 °C. Following this, ATP was added at 2.5 mM final concentration, prepared at 25 mM stock pH adjusted to 7.0 with 3% KHCO_3_ 3 M. Samples were preincubated for 30 s and then the absorbance monitored for 2 min. At that time oligomycin was added to a final concentration of 2 µg/mL. Calculations: Absorbance_(340nm no oligomycin)_ − Absorbance _(340nm with oligomycin_)* volume (mL)/ε(=6.22 mL/nmol*cm)* sample volume (mL)* protein concentration in mg/mL*1 cm light path.

### 4.12. Blue Native Gel Electrophoresis

Native PAGE was carried out as previously reported [51]. Briefly, 3–13% acrylamide gradient gels were loaded with 60 μg of mitochondrial protein and processed as described. After electrophoresis, proteins were transferred to a PROTAN^®^ nitrocellulose membrane (Schleicher and Schuell, Bioscience GmbH, Dassel, Germany) at 35 V, overnight, and probed with specific antibodies raised against the following human OXPHOS subunits: NDUFA9, CORE2, COX2, COX5A, and SDHA (from Mitosciences, Eugene, OR, USA). Peroxidase-conjugated anti-mouse IgG was used as a secondary antibody (Molecular Probes, Thermofisher Scientific, Gilligham, UK). The signal was detected with ECL^®^ plus (Amersham Biosciences, Freiburg, Germany).

### 4.13. Proteome Analysis

Cells were cultured in SILAC DMEM media (Thermo Fisher) supplemented with dialyzed FBS, 100-units/mL penicillin/streptomycin, 2 mM L-glutamine and arginine (R) and lysine (K) isotopes to a final concentration of 84 mg/L and 146 mg/L, respectively, as follows: [^12^C]_6_[^14^N]_4_-L-Arg and [^12^C]_6_[^14^N]_2_-L-Lys (Merck-Sigma-Aldrich L9037 and A6969) were used for R0K0 ‘light’ (L) medium to culture C2C12 CBL KD and NTshRNA-expressing cells. ‘Heavy’ isotope aminoacids, U-[^13^C]_6_[^15^N]_4_-L-Arg (Thermofisher, 89990) and [^13^C]_6_[^14^N]_2_-L-Lys (Merck-Sigma-Aldrich, 603041), were used for R10K6 ‘heavy’ medium to culture C2C12 untransfected control cells. The incorporation of the heavy amino acids was confirmed by MS after six cell doublings. For the experiment, cells were differentiated for 4 days lysed in water with a brief sonication and crude mitochondrial fractions obtained by centrifugation as indicated above. Equal protein amounts (10 µg) of each pair of ‘heavy’ and ‘light’, in triplicate samples, were mixed 1:1 resolved by SDS-PAGE, stained in Coomassie Blue and sliced. Each slice was subjected to in-gel tryptic digestion and peptide extraction followed by LC-MS/MS analysis in a Thermo LTQ-OrbitrapXL LC-MS mass spectrometer equipped with a nanospray source and interfaced to a Waters nanoAcquity UPLC system. MS data were analyzed using Andromeda and MaxQuant 1.6.3.4 against a reviewed mouse UniProt protein database. Functional analysis and data visualization were carried out in R/Bioconductor v.3.4.

### 4.14. Statistical Analysis

Data are presented as mean ± SEM. Comparisons between treatment groups were performed by Student’s *t*-test or ANOVA (GraphPad Prism, GraphPad Software Inc., San Diego, CA, USA) as indicated. *p* < 0.05 was considered statistically significant; * *p* < 0.05; ** *p* < 0.01; *** *p* < 0.001.

## Figures and Tables

**Figure 1 ijms-24-03399-f001:**
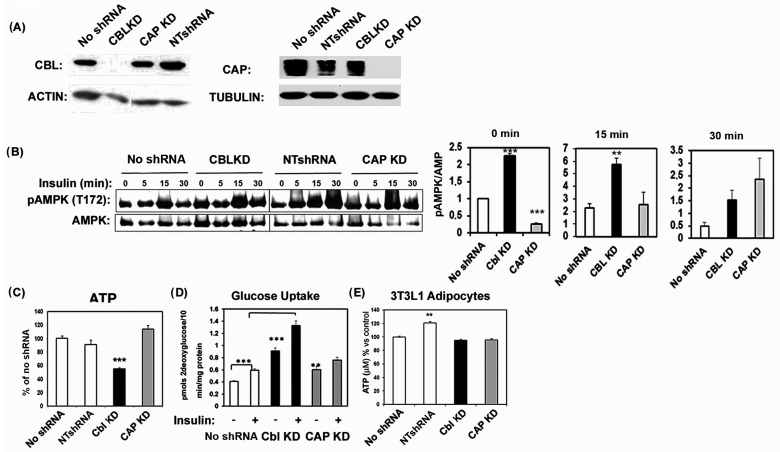
CBL depletion in C2C12 myotubes decreases ATP levels and enhances AMPK activity and glucose transport. (**A**) Western blot analysis showing knock down of Cbl and CAP in C2C12 myotubes. Whole cell lysates obtained from untransfected cells (None) or from cells expressing shRNAS for CBL (CBL KD), or CAP (CAP KD), or non-targeting shRNAs (NTshRNA) were immmunoblotted with specific antibodies as indicated (**B**) Activation of AMP regulated kinase (AMPK) in CBL KD, CAP KD, and control cells. Differentiated myotubes were serum starved and either left untreated (0 min) or treated with 100 nM insulin for 15 or 30 min Cellular lysates were obtained, separated by SDS-PAGE and immunoblotted with antibodies for phospho-AMPK (T172) and total AMPK. The right panel shows the quantification of normalized phospho-AMPK from Western blot data from *n* = 3 experiments at 0, 15, and 30 min post-treatment with 100 nM insulin. Quantification was carried out with Image J, NIH. Graphs show mean ± SEM values. Statistical analysis: one-way ANOVA to no shRNA expressing cells. ** indicates *p* < 0.01; *** *p* < 0.001. (**C**) ATP content in cellular lysates of differentiated C2C12 cells expressing no shRNAs, NTshRNAs or shRNAs specific for CBL or CAP. Graphs show mean ± SEM values of data combined from 3 experiments each with *n* = 3–4 biological replicates per group. ATP values are expressed as % of values in control cells. Statistical analysis: one-way ANOVA. *** indicates *p* < 0.001. (**D**) ^3^H-2-deoxyglucose transport in C2C12 myotubes. Uptake was measured in basal (non-stimulated) conditions and following the stimulation of insulin (100 nM, 30 min) No shRNA-expressing cells or cells expressing shRNAs for CBL or CAP genes as indicated. Data show a mean ± SEM of a representative experiment with *n* = 4 biological replicates. Statistical analysis: one-way ANOVA to NTshRNA-expressing cells. ** indicates *p* < 0.01; *** *p* < 0.001. (**E**) ATP cellular content in differentiated 3T3L1 adipocyte cells expressing no shRNA, NTshRNA or shRNAs specific for CBL or CAP. Data show mean ± SEM of *n* = 4. ** indicates *p* < 0.01 compared to control (No shRNA).

**Figure 2 ijms-24-03399-f002:**
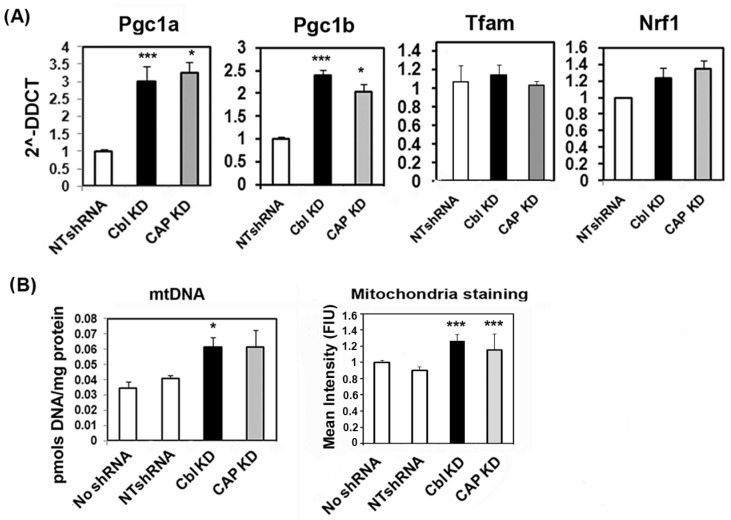
CBL or CAP depletion increases mitochondrial mass. (**A**) Quantification of mRNA of genes involved in mitochondrial biogenesis. Total RNA was obtained from C2C12 myotubes expressing NTshRNA or shRNAs for CBL or CAP. mRNA for each gene was determined by qPCR. Data show the relative expression to control cells; graphs are mean ± SEM values of *n* = 3–9 samples per group. (**B**) Assessment of mitochondrial mass. Quantification of mtDNA by qPCR (left graph) and quantitation of fluorescently labelled mitochondria with monomeric JC-1, arbitrary units (right graph). Graphs shows mean ± SEM of data from 5 experiments, each performed in triplicate biological samples. Statistical analysis: one-way ANOVA to NTshRNA-expressing cells. * indicates *p* < 0.05, and *** *p* < 0.001.

**Figure 3 ijms-24-03399-f003:**
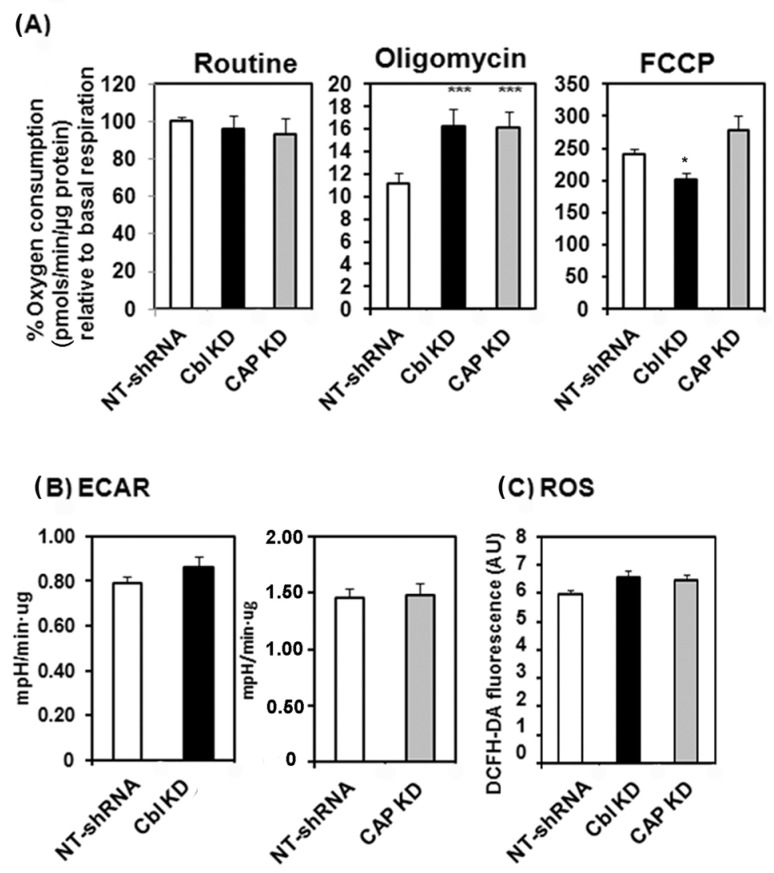
CBL- and CAP-depleted myocytes show alterations in mitochondrial respiration. (**A**) Oxygen consumption in CBL and CAP knockdown cells compared to control cells expressing NT-shRNA. The data show the mean ±SEM of a representative experiment. Routine: basal respiration, FCCP 0.5 μM, oligomycin 2.5 μM concentrations. Statistical analysis: one-way ANOVA to NTshRNA-expressing cells. * indicates *p* < 0.05, *** indicates *p* < 0.001. (**B**) Extracellular acidification rates (ECAR) values, data show mean ±SEM of a representative experiment. Not statistically significant. (**C**) Reactive oxygen species. Cell mitochondria were stained with DCFH-DA as specified in Materials and Methods and emission fluorescence quantified at 560 nm. Graphs show mean ±SEM values of fluorescence intensity (arbitrary units) of *n* = 163 control, *n* = 126 CBL KD, and *n* = 192 CAP KD cells. Statistics: one-way ANOVA, not statistically significant.

**Figure 4 ijms-24-03399-f004:**
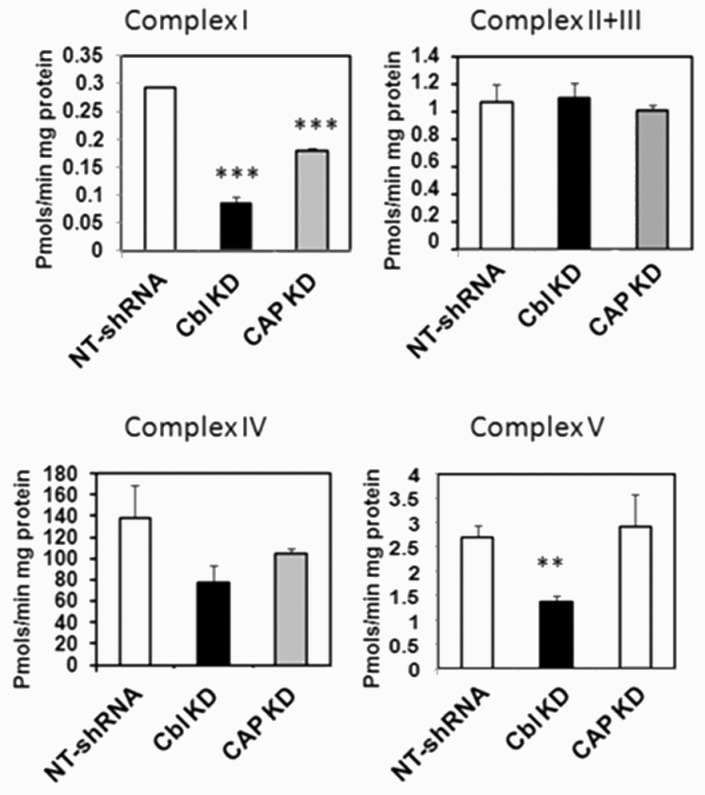
Mitochondrial respiratory complex I activity is reduced in CBL KD. Mitochondria were isolated by differential centrifugation. The activities of OXPHOS complexes were determined in vitro using spectrophotometric methods under non-limiting substrate conditions as explained in the section Materials and Methods. Data for each respiratory complex are expressed as nmol/min/mg or pmol/min/mg protein relative to NTshRNA-expressing cells. Graphs show mean ± SEM of 3 independent experiments. Statistical analysis: one-way ANOVA to NTshRNA-expressing cells. ** indicates *p* < 0.01; *** *p* < 0.001.

**Figure 5 ijms-24-03399-f005:**
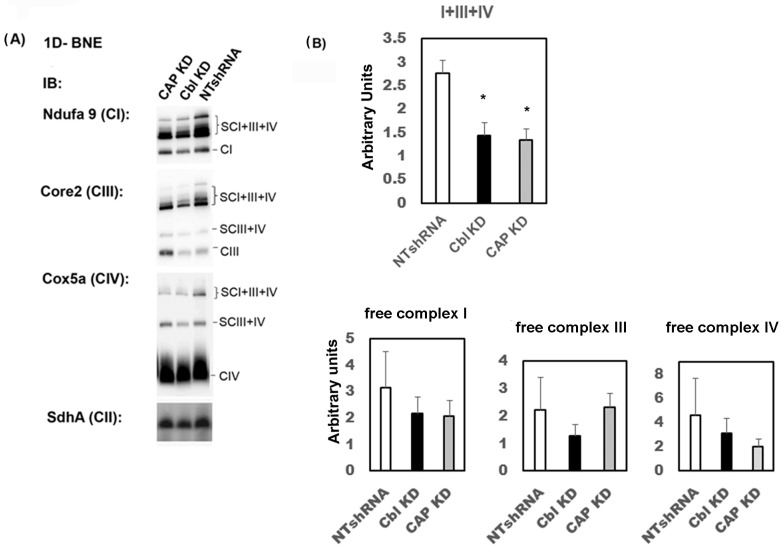
Reduced mitochondrial respirasome assembly in CBL- or CAP-depleted cells. Mitochondria were isolated from C2C12 control or cells expressing shRNAs for c-CBL (CBL KD) or CAP (CAP KD) or NTshRNAs. (**A**) Proteins were separated by 1D-BNE and immunoblotted with antibodies against NDUFS9 (complex I), CORE 2 (complex III), COX5a (complex IV) and SDHA (complex II). Each OXPHOS complex is indicated in brackets. Immunoblots show a representative experiment of *n* = 4 independent experiments. (**B**) Quantification of the relative abundance of individual respiratory chain complexes and assemblies, supercomplexes, and individual complexes normalized to SDHA (CII subunit). Western blots for 4 independent experiments were quantitated in Image J (NIH). Values for free complexes (lower graphs) were calculated by normalizing the densitometric signals of bands CI, CIII, and CIV by those of SDHA. Values for supercomplexes (upper graph) were obtained by calculating the mean of the densitometric signals of band SCI + III + IV obtained with antibodies NDUFS9, CORE2, and COX5a- for each independent experiment, further normalized by CII (SDHA signals). Graphs show mean ± SEM values. Statistical analysis: one-way ANOVA to NTshRNA-expressing cells. * indicates *p* < 0.05.

**Figure 6 ijms-24-03399-f006:**
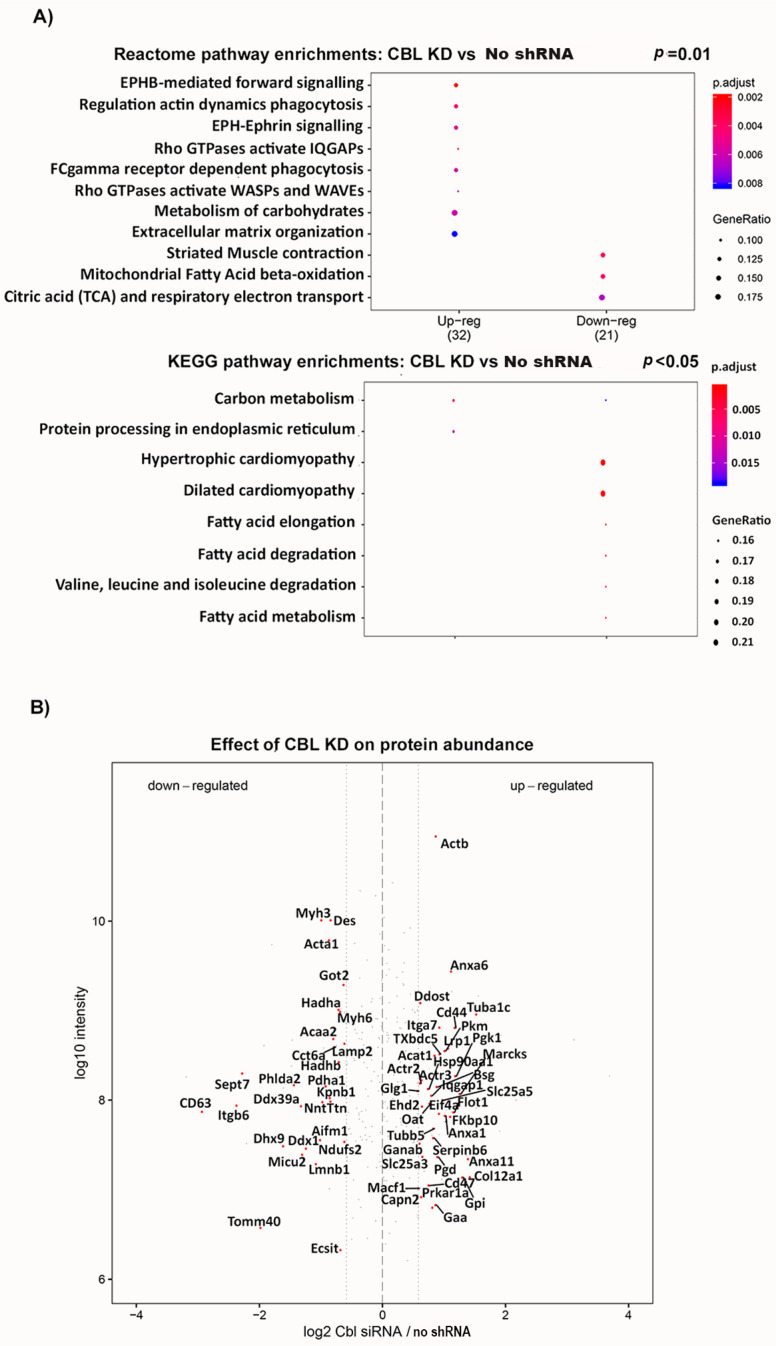
Cbl KD cells have reduced abundance of proteins involved in CI biogenesis and fatty acid degradation, and increased levels of glycolytic enzymes. SILAC-based proteome comparison of CBL KD to untreated control cells. Functional analysis and visualization of data were carried out in R/Bioconductor package v.3.4. All protein hits from the 1% FDR Andromeda search were subsequently used for GO enrichment analysis. (**A**) Reactome and KEGG pathway term enrichment analysis of differentially expressed proteins. (**B**) Relative abundance of differentially expressed proteins. Those highlighted in red show 1.5-fold or greater differential expression.

**Figure 7 ijms-24-03399-f007:**
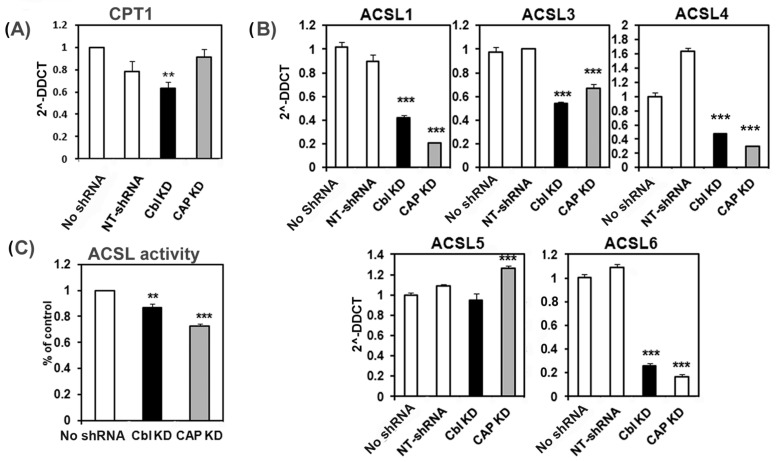
CBL- and CAP-depleted cells have reduced expression of genes involved in fatty acyl-CoA import into mitochondria. mRNA levels of *Cpt1* (**A**) or long chain Acyl CoA synthases (*Acsl1-6*) (**B**) were determined by qPCR. (**C**) Acetyl CoA liase activity was measured on ^14^C-palmitate in cellular lysates from untransfected (no shRNA), CBL shRNA (CBL KD), CAP shRNA (CAP KD), or NTshRNA-expressing cells. Graphs show mean ± SEM. Statistical analysis: one-way ANOVA to control untransfected with Bonferroni post hoc test. ** indicates *p* < 0.01, *** *p* < 0.001.

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
