# Peer review of "CBL/CAP Is Essential for Mitochondria Respiration Complex I Assembly and Bioenergetics Efficiency in Muscle Cells"

_ijms, 2023, doi:10.3390/ijms24043399_

Round 1

Reviewer 1 Report

The authors present a study to explore the possible mechanisms through which the CBL regulates mitochondrial mass and function in C2C12 myocytes. The study presents consistent and relevant results. However, there is a major issue of how Cbl is exerting its effects on mitochondrial respiratory chain complex I biogenesis that remains unclear.

1.  The Cbl family comprises three distinct mammalian members: c-Cbl, Cbl-b and Cbl-3, why is c-Cbl selected for study?

2.   Fatty acid oxidation is directly related to the mitochondrial function. CBL KD cells exhibited elevated AMPK activity, which has been shown to increase fatty acid oxidation. It is necessary to characterize the fatty acid oxidation in these cells.

3.   The authors show lower expression of several proteins involved in fatty acid oxidation in this in vitro study. However, the Cbl knockout mice showed increased fatty acid oxidation lower plasma FFA levels (Ref. 27). Can the authors explain this discrepancy?

4.    In figure 1b, quantification of the pAMPK should be provided.

Reviewer 2 Report

The authors use a wide spectrum of techniques to characterize changes in a model muscle system (cultured myocytes) associated with shRNA knockdown of proteins (CBL and associated protein CAP) in the insulin signaling pathway.  While it is mostly descriptive (mechanisms not defined) it is nonetheless very interesting and points to targets for further study. However, while the overall conclusions are mostly sound, there are technical issues that cloud some results and moderately reduce enthusiasm (listed below). These should be readily addressable by the authors (even as disclaimers) prior to publication. 

1. Clearly the main emphases of the authors are defects in respiratory chain Complex I, decreased fatty acid beta oxidation, and a shift from oxidative metabolism to glycolysis associated with insulin signaling. This raises an important question about how good a model system the C2C12 myocytes are, since cell lines tend to be more glycolytic than primary cell lines. For example, are the C2C12 basal respiration rates (per mg cell protein) and ratio of ATP provided by oxphos and glycolysis comparable to freshly isolated or cultured myocytes?  Results may need to be framed by these considerations.

2. The controls in experiments are erratic. In some cases both NT shRNA and untreated cells are used. In others one or the other. (The label "None" is used in several figures but undefined.) This is a concern since the control proteomic experiment (in supplemental material) seem to indicate that perhaps 10%+ of proteins studied vary significantly between the two control conditions. Can the authors explain their use of controls in different experiments and provide assurance that the significance of results did not vary significantly based on which control was used?  

3. In the quantification of western results in Fig 5B, what antibody was used for SCI+III+IV? Also, the lack of significance for the changes in free complexes I, III, IV seems to be due to large variances in the NT shRNA samples. Were the NT shRNA results generally more variable? (possibly related to question 1)?

3. Provide the p value ranges associated with the appropriate symbols in the figures. In Fig 5 no definition is provided for *. In Figs 2 and 3, ** is defined but the symbol is *.  

4. Bioenergeticists such as this reviewer would be interested in the findings relative to ATP synthase, i.e., specific activity is reduced (Fig 4D) and oligomycin-inhibited respiration is increased. However, when I tried to find the protocol for ATP synthesis (as well as Complexes I-IV), Methods said fully described in Supplemental Material but it was not found. The reference for these protocols is apparently a book that is behind a paywall for this reviewer. Please provide details for the assays. 

5. Uptake of JC-1 is dependent on inner membrane potential and thus is not usually considered appropriate for measuring mito mass. Since knockdowns are associated with changes in oxphos, this would seem to be a legitimate concern. Can the authors justify use of JC-1 based on their own experience or that of others (perhaps there is no significant difference in basal IM potential for controls and knockdowns). Alternatively, they could provide a disclaimer that the mito mass result might be inaccurate for this reason (I don't think the result is critical to their bottom line.)  

Author Response

Please see attached file with point-by-point response.

Round 2

Reviewer 1 Report

It would be nice if the authors discuss more about the mechanisms by which Cbl  regulates the biogenesis of mitochondrial respiratory chain complex I.
